# Mechanical Behavior of Repaired Monolithic Crowns: A 3D Finite Element Analysis

**DOI:** 10.3390/dj11110254

**Published:** 2023-10-31

**Authors:** Pablo Machado Soares, Lucas Saldanha da Rosa, Gabriel Kalil Rocha Pereira, Luiz Felipe Valandro, Marilia Pivetta Rippe, Amanda Maria de Oliveira Dal Piva, Albert J. Feilzer, Cornelis J. Kleverlaan, João Paulo Mendes Tribst

**Affiliations:** 1Post-Graduate Program in Oral Sciences, Center for Development of Advanced Materials, Division of Prosthodontics-Biomaterials, Federal University of Santa Maria (UFSM), Santa Maria 97105-900, Brazillucas.saldanha@acad.ufsm.br (L.S.d.R.); gabriel.pereira@ufsm.br (G.K.R.P.);; 2Department of Dental Materials Science, Academic Centre for Dentistry Amsterdam (ACTA), Universiteit van Amsterdam and Vrije Universiteit Amsterdam, 1081 LA Amsterdam, The Netherlands; a.m.de.oliveira.dal.piva@acta.nl (A.M.d.O.D.P.);; 3Department of Reconstructive Oral Care, Academic Centre for Dentistry Amsterdam (ACTA), Universiteit van Amsterdam and Vrije Universiteit Amsterdam, 1081 LA Amsterdam, The Netherlands; a.feilzer@acta.nl

**Keywords:** ceramic restoration, glass–ceramic, stress distribution, shear bond, zirconium, finite element analysis, resin composite

## Abstract

This study evaluated the mechanical behavior and risk of failure of three CAD-CAM crowns repaired with different resin composites through a three-dimensional (3D) finite element analysis. Three-dimensional models of different cusp-repaired (conventional nanohybrid, bulk-fill, and flowable resin composites) crowns made of zirconia, lithium disilicate, and CAD-CAM resin composite were designed, fixed at the cervical level, and loaded in 100 N at the working cusps, including the repaired one. The models were analyzed to determine the Maximum Principal and Maximum Shear stresses (MPa). Complementary, an in vitro shear bond strength test (*n* = 10) was performed to calculate the risk of failure for each experimental group. The stress distribution among the models was similar when considering the same restorative material. The crown material affected the stress concentration, which was higher for the ceramic models (±9 MPa for shear stress; ±3 MPa for tensile stress) than for the CAD-CAM composite (±7 MPa for shear stress; ±2 MPa for tensile stress). The shear bond strength was higher for the repaired CAD-CAM resin composite (±17 MPa) when compared to the ceramics (below 12 MPa for all groups), while the repair materials showed similar behavior for each substrate. The stress distribution is more homogenous for repaired resin composite crowns, and a flowable direct resin composite seems suitable to repair ceramic crowns with less risk of failure.

## 1. Introduction

The use of computer-aided design computer-aided manufacturing (CAD-CAM) to produce metal-free restorations through different approaches such as subtractive (milling) and additive (printing) manufacturing techniques has increased exponentially in the last few years [1]. Digital dentistry allowed the acquisition of monolithic restorations with fewer defects compared to bilayer crowns. Additionally, it offers heightened accuracy, all within a single clinical appointment [1]. In addition, metal-free restorations promote a more conservative tooth preparation when compared to metal ceramics [2], making it a viable and health-conscious option for oral rehabilitation.

In the evolving field of dentistry, the range of choices for monolithic restorations has expanded considerably. With advancements in materials and technology, dental professionals now have a wide range of options to choose from, each catering to specific clinical needs and patient preferences. Lithium disilicate is a widely used material for such purposes, as it offers exceptional mechanical properties and aesthetic appeal. This is due to its unique microstructure, which contains a silica matrix and a high amount of crystal phase [3]. Alternatively, zirconia is also widely used, since it presents the highest mechanical strength among ceramic materials [4], and advances in the manufacturing of high-translucent zirconia (increase in yttrium stabilizer and cubic phase) made this material suitable for both anterior and posterior restorations [5]. Even so, despite the high reported survival rates for monolithic ceramic restorations in 3 years (97.0% for monolithic glass–ceramic restorations; 96.1% for monolithic zirconia), small chipping and fractures were also reported [6]. In the search for overcoming the inherent brittleness of traditional crowns and ensuring a consistent stress distribution, the utilization of materials mimicking Young’s modulus of dentin has emerged as a fundamental strategy in the field of dental materials. Among these materials, CAD-CAM resin composites have emerged as a promising alternative, gaining significant popularity among dental professionals [7,8]. What sets these composites apart is their remarkable combination of mechanical strength and aesthetic characteristics. Studies have consistently shown promising mechanical behavior, indicating their structural integrity and durability under various stress conditions [8]. Moreover, these resin composites exhibit excellent optical properties, which can be properly merged with natural teeth to enhance the overall aesthetic appearance of dental restorations [7]. This dual advantage not only highlights their unique properties but also positions them as a versatile solution in modern dentistry, offering not just reliable restorations but also a superior visual appeal.

When chipping occurs in dental crowns, clinicians face a crucial choice: whether to replace the crown entirely or opt for a conservative approach. This decision hinges on factors such as the extent of the damage, its location, and the patient’s overall oral health. Clinicians must carefully balance preserving the existing restoration with ensuring the long-term satisfaction and oral health of the patient. The choice underscores the complexity of dental care, emphasizing the need for personalized approaches adapted to each patient’s unique circumstances. According to Heintze and Rousson [9], the alternatives include the polishing of the fractured portion of the restoration (grade 1), the repair of the restoration with a resin composite (grade 2), and total replacement with a new crown (grade 3). In comparison with grade 3, the grade 2 repairs with a resin composite are cheaper, less invasive, and can be performed in one section. Also, previous studies reported that repair protocols consist of an efficient alternative for the longevity of restorations through satisfactory mechanical behavior, even though it is not as high as non-fractured crowns [10,11]. In this sense, different materials are indicated as options to perform the repair, including different types of resin composite [12,13]. Conventional nanohybrid composites have been widely used for oral restorations, including for repair, since they are aesthetic and promote low polymerization shrinkage [14,15].

As alternatives to the nanohybrid materials, a bulk-fill resin composite has been suggested for such approaches, since it simplifies the technique, allowing bigger increments than 2 mm with efficient light activation [13,15,16]. Akgül et al. [13] compared the use of both bulk-fill and conventional resin composite to repair resin-based restorations and reported that there were no differences between them for the bond strength values. In addition, the flowable resin has also been reported for repair [12,17], due to the possibility of obtaining a more well-filled interface with the indirect restoration. Nevertheless, there is a notable gap in understanding the mechanical performance of repaired CAD-CAM restorations, especially concerning the application of bulk-fill and flowable resin composites, particularly in the context of both ceramic and resin-based indirect restorations. Addressing this gap is crucial. In this regard, in silico studies emerge as a vital alternative. They provide valuable insights into stress distribution (using numerical analysis) when employing these materials, offering essential guidance for selecting the appropriate repair composite in diverse clinical scenarios.

Therefore, considering the aforementioned factors, the present study aimed to evaluate the mechanical behavior and risk of failure of different CAD-CAM materials (lithium disilicate, translucent zirconia, and resin composite) repaired with three classes of a resin composite (nanohybrid, bulk-fill, and flowable resin), through a three-dimensional (3D) finite element analysis. The null hypothesis was that the stress distribution and measured shear bond strength would not be affected by (1) the CAD-CAM indirect material and (2) the repair resin composite.

## 2. Materials and Methods

### 2.1. Finite Element Analysis (FEA)

Three-dimensional models of different composite-repaired (Tetric Evoceram, Tetric PowerFlow, and Tetric PowerFill, Ivoclar AG, Schaan, Liechtenstein) crowns made of zirconia (Yz—IPS e.max ZirCAD MT, Ivoclar AG), lithium disilicate (Ld—IPS emax CAD, Ivoclar AG), and CAD-CAM resin composite (Tc- Tetric CAD, Ivoclar AG) were designed in the modeling software (Rhinoceros version 5.0 SR8, 2013, McNeel North America, Seattle, WA, USA). The first molar crown model was described in a previous study [18], imported, and then modified into CAD software to present a cusp-repaired monolithic crown, according to each evaluated material. Different views of the models and the meshing are depicted in Figure 1.

The models were exported to computer-aided engineering (CAE) software (ANSYS 19.0, 2018, ANSYS Inc., Houston, TX, USA). To ensure the accuracy of our results, a rigorous 10% mesh convergence test was conducted for validation purposes. This numerical approach not only validated the results but also facilitated the quantitative evaluation of stress concentration differences between the various groups. A data analysis was performed using computer-aided engineering software. The Young modulus of each restorative and repair material [8,19,20,21] was used for each solid considering an isotropic and linear behavior (Table 1). The contacts were considered perfectly bonded. After the meshing process, the model was fixed at the cervical level, and a standardized load of 100 N was applied at the working cusps, including the repaired one. The Maximum Principal and Shear stresses (MPa) were measured in each model.

### 2.2. In Vitro Shear Bond Strength Test

To determine the risk of failure for each group, an in vitro shear bond strength test was carried out. Fifteen discs were obtained from each restorative material (TZ—IPS e.max ZirCAD MT, Ivoclar AG; LD—IPS emax CAD, Ivoclar AG; RC—Tetric CAD, Ivoclar AG) from CAD-CAM blocks according to methodological steps described in previous studies [8,20]. The blocks were turned into cylinders (Ø = 10 mm) with the use of a grinding machine (ECOMET Grinder/Polisher, Buehler, Lake Bluff, IL, USA) first with a diamond grinding disc (Dia-Grid Diamond Discs #120—average grit size: 160 μm, Allied High Tech Products, Inc. Rancho Dominguez, Los Angeles, CA, USA) and then with Silicon Carbide (SiC) papers in the sequence of #400 and #600 grit size. Then, the cylinders were cut into discs in a precision cutting machine with a diamond saw (Isomet 1000, Buehler), under constant water irrigation. The obtained discs were ground with the use of #200, #400, #600, and #1200 SiC papers on both sides until reaching a polished surface. Translucent zirconia and lithium disilicate were then fired in specific furnaces according to the manufacturers’ recommendations, thus achieving the desired final dimensions for all materials (Ø = 10 mm and 1.5 mm of thickness).

Before the application of the repair resin composites, different surface treatments were performed on the bonding surface of the restorative specimens according to the manufacturer’s instructions for each material. Translucent zirconia and CAD-CAM resin composite discs were air-abraded with alumina particles (50 µm gran size) for 10s at 2 bar of pressure [8,20]. A 10-metacriloiloxidecil di-hydrogenic phosphate (10-MDP) primer (Alloy Primer, Kuraray Noritake) was then applied over the zirconia surface for 10 s, followed by a gentle air drying. A universal adhesive (Adhese Universal, Ivoclar AG) was applied over the CAD-CAM resin composite for 20 s and then light activated at 1200 mW/cm^2^ (Radii-cal LED curing light, SDI, Bayswater, Australia) for 10 s.

The lithium disilicate glass–ceramic was treated with 5% hydrofluoric acid etching (IPS ceramic etching-gel, Ivoclar AG) for 20 s, and then washed in running water for 30 s. A silane-based coupling agent (Monobond N, Ivoclar AG) was actively applied for 15 s, allowed to react for another 45 s, and then gently air-dried for 10 s [20].

The treated ceramic discs were randomly distributed and the direct repair resin composites were then applied with the increment technique over the adhesive surface (2 cylinders per restorative disc) with the use of a cylindric matrix (Ø = 3.20 mm) according to each group (*n* = 10; experimental unit: resin composite cylinders) and then light cured for 20 s. The specimens were stored in distilled water for 24 h at 37 °C before the shear bond strength test. The test was performed in a universal machine (Instron 6022; Instron, Norwood, MA, USA) at a crosshead speed of 1 mm/min. The specimens were positioned vertically in a metal base, and the load (1 KN load cell) was applied at the interface between the composite cylinders and the restorative discs using a flat stainless steel load applicator (Ø = 10 mm). The load for failure data was collected, and the shear bond strength (S) was calculated by using the formula S = F/A, where “F” is the load for failure and “A” is the interface cylindric area (8.04 mm^3^).

### 2.3. Statistical Analysis

Shapiro Wilk and Levene tests were performed to assess the normality and homoscedasticity of the obtained data, respectively. The in vitro shear bond strength test data were analyzed under two-way ANOVA and Tukey post hoc testing (α = 0.05) considering the restorative substrate and the repair resin composites factors.

### 2.4. Risk of Failure Measurement

After the determination of the Maximum Shear Stress and shear bond strength for each repaired restoration (MPa), the risk for failure was calculated according to the following formula: shear stress peak/shear bond strength [22].

## 3. Results

The mean results of Maximum Principal and Shear stresses are depicted in Table 2. Figure 2 shows the model’s deformation according to each generated stress, restorative material, and repair composite.

### 3.1. Maximum Tensile and Shear Stresses

The Maximum Shear Stress generated in the models was higher than the Maximum Principal stress, regardless of the evaluated repaired material. In addition, the stress distribution among the repair materials was similar when considering the same restoration model (Table 2).

The crown material affected the results, with the tensile and shear stresses being 36.6% and 17.1% higher for the ceramic models (zirconia and lithium disilicate) than for the CAD-CAM composite model, respectively. Also, the CAD-CAM composite crown suffered more deformation during the load application, regardless of the repair material (Figure 2).

### 3.2. In Vitro Shear Bond Strength Test

The in vitro shear bond strength test is also depicted in Table 2. Two-way ANOVA indicated that while the repair resin composite did not affect shear bond strength (*p* = 0.13; F = 2.08), the substrate affected the results (*p* = 0.00; F = 21.47). Furthermore, the highest bond strength values were observed for the repaired CAD-CAM resin composite material, which were similar to zirconia and lithium disilicate repaired with a flow resin composite (*p* > 0.05) and higher (*p* < 0.05) than zirconia and lithium disilicate both repaired with nanohybrid and bulk-fill resin composites.

### 3.3. Risk of Failure

Regarding the risk of failure, the translucent zirconia and lithium disilicate crowns repaired with bulk-fill resin composites presented the highest percentages of risk and trespassed the survival limit (135% and 124% risk, respectively), followed by TZ and LD repaired with a nanohybrid resin composite (112% and 102% risk, respectively). Therefore, it is expected that lithium disilicate and translucent zirconia ceramics repaired with nanohybrid and bulk-fill resin composites would fail after 100 N of load application. CAD-CAM-resin-composite-repaired crowns showed the lowest risk of failure for all repair materials (Figure 3).

## 4. Discussion

The success of repair procedures depends on achieving a strong and consistent bonding between the repair material and the damaged restoration, the same as the homogenous mechanical behavior of both substrates [23]. In this sense, the Finite Element Analysis is an essential tool to evaluate the stress distribution along the adhesive interface, to assist in determining the best approach in terms of material choice for clinical applications. According to the results of the present study, the repair material did not affect the mechanical performance of the restorations. However, the CAD-CAM indirect material to be repaired affected the stress distribution in the models. Thus, the null hypothesis was partially rejected.

Ceramic materials are widely recommended for oral rehabilitations, considering their excellent mechanical behavior, and biocompatibility, besides being more resistant to wear and pigmentation when compared to resin composites [24,25,26]. Even so, monolithic ceramic restorations are also susceptible to cracks and fractures, mainly when considering a fatigue stimulus [27]. In those cases, the direct repair with a resin composite is indicated to return the function and aesthetic of the crown, also being a more conservative approach when compared to the replacement of the entire restoration [28]. The evaluation of the stress distribution showed that both shear and tensile stress were higher for the zirconia and lithium disilicate when compared to the CAD-CAM resin composite. This may be explained by the Young modulus of the ceramic materials (200 GPa for zirconia; 95 GPa for lithium disilicate) that is much higher than the direct resin composite (11 GPa) [8,29], resulting in higher stresses and almost no deformation in the zirconia and lithium disilicate (Figure 2), as illustrated with the stress peaks for the evaluated models (Figure 2). These findings follow previous studies, which depicted higher stress concentration within more rigid materials [30] such as dental ceramics. As a consequence, less stress was transmitted to the interface and the repair. In addition, zirconia and lithium disilicate present the highest mechanical strength among the ceramic materials [20], which makes the difference even higher when comparing them to the composite. There was no difference in the stress distribution between zirconia and lithium disilicate. Even though zirconia presents a higher modulus than the glass–ceramic [20], both are considered stiffer when compared to the repair composite; thus, the difference was probably not enough to generate a significant discrepancy in terms of stress concentration.

When the repaired CAD-CAM resin composite was loaded, the tensile and shear stresses were lower within the crown, regardless of the repair material. The reduced tensile and shear stresses within the crown indicate an effective distribution of loads and an enhanced ability of the repair material to reinforce the structural integrity of the composite. CAD-CAM composites consist of materials that are polymerized under higher pressures and temperatures than direct resin composites, thus generating a restorative material with improved mechanical properties when compared to conventional resin composites [31]. However, the Young modulus of the evaluated CAD-CAM indirect resin composite is closer to the repair material (Table 1); thus, the generated stresses were more homogenous, and a lower magnitude was observed between the direct resin and the crown (Table 2), which decreases the risk of interfacial failure for the repair (Figure 3). In addition, the resin composite model underwent a greater deformation than the ceramic models (Figure 2), probably due to the microstructure of the composite, which is susceptible to plastic deformation and is less rigid than zirconia and lithium disilicate [8,21]. The bonding potential between resin-based materials is also another factor to be considered since it is much higher due to the similarity of both composites and high bond strength values were previously reported when repairing resin composite restorations [32]. Hence, it may be expected that a stronger and more stable interface is achieved when using only polymeric materials, with less chance of debonding when the restoration is under stress during the load application.

The material of choice to perform the repair procedure was also evaluated in the present study. Direct repairs are indicated as a simple and less time-consuming approach than the total replacement of the crown. Several brands and types of resin composite have been used for such protocols, including nanohybrid, micro-hybrid, bulk-fill, and flowable composites [12,13,15,16]. Taking into account the nano-sized filler particles, nanohybrid composites are one of the most commonly used materials for repair, since they provide great aesthetics, durability, and low polymerization shrinkage [15,33]. Also, the flowable composite was previously evaluated for repair procedures [12], due to its capacity to fill surface defects and generate a more stable interface, while the bulk-fill one has been recommended to fill deeper cavities in one increment [13]. However, the present research findings indicate that all repair materials exhibited similar stress distribution patterns, irrespective of the specific CAD-CAM material they were intended to repair. This consistency was observed even when comparing composites with varying filler content and sizes, all of which displayed low elastic modulus values (as shown in Table 1). This shared characteristic likely accounts for the uniform stress concentration observed across all models that were simulated. Therefore, it can be assumed that stress homogeneity is maintained within the system when focusing solely on resin-based materials.

Despite showing similar mechanical responses when loaded, the different resin composites present different compositions and bondability with ceramic materials. In this sense, resin composites with higher bond strength withstand more stress at the adhesive interface than the ones that present a weaker link to the substrate. The in vitro shear bond strength test showed that the values between the repair composites and ceramic materials range from approximately 7 MPa to 18 MPa, with the shear bond strength values being higher for CAD-CAM resin composite crowns. Even so, when repairing ceramic restorations with a flowable resin composite, the bond strength values were similar to those achieved for the CAD-CAM composite restoration (Table 2). A flowable resin composite presents more monomers and less viscosity due to its lower filler content and smaller size of particles, thus providing more probability to adapt on a bonding surface when compared to other conventional resin composite materials, with promising results in terms of bond strength showed in previous studies [34,35,36]. As a result, the adhesive interface may exhibit more contacting surfaces and consequently, a greater resilience to stress when repairing monolithic ceramic crowns using the flowable resin composites. Thus, a lower risk of failure may be expected in this scenario, as corroborated by the findings of the present study (Table 2, Figure 3).

A finite element analysis (FEA) has revolutionized dentistry by providing an accurate and efficient method to evaluate various aspects of oral healthcare [8,18,20,22]. In the scope of restorative dentistry, numerical simulations are instrumental in optimizing the design of dental cavities, ensuring their durability and success within the dynamic oral environment. FEA also guides the development of dental materials, enabling the creation of bio-inspired and long-lasting prosthetics like crowns and bridges [29,30]. Moreover, FEA facilitates the customization of treatments, accurately predicting tooth movement patterns as well as stress and strain distribution [18]. Additionally, in complex oral and maxillofacial surgeries, FEA assists dentists in planning complex procedures, improving outcomes by minimizing risks, and enhancing the effectiveness of different techniques [36,37,38,39,40].

Despite all the advantages, as an in silico study, the limitations must be appointed. Just one geometry of the repaired crown was evaluated, considering only a single cusp repair. In addition, surface treatments and conditioning methods were not factors in the present study; therefore, the evaluation of the interface between the crown and the repair material must be considered with caution. Each protocol was used according to the microstructure and manufacturer’s recommendations according to the crown material, aiming to mimic the clinical scenario. It is important to consider that other factors are determinants for the clinical longevity of repaired restorations, such as the challenges of the oral environment, pH variations, parafunctional habits, and multidirectional load applications. Nevertheless, it is crucial to underscore the significance of finite element analysis (FEA) studies in comprehending the mechanical dynamics of restorative configurations within a standardized and controlled environment, to evaluate the stress concentration as an isolated factor. Consequently, FEA serves as a valuable tool to forecast the performance of repair protocols for distinct restorative materials, which is essential to obtain a definitive repair protocol for repair procedures.

## 5. Conclusions

The stress distribution diverged according to crown material, with resin composite crown repairs displaying more even and diminished stress, while ceramic crowns exhibited heightened stress concentration. Repairing translucent zirconia posed a greater risk of failure compared to lithium disilicate and CAD-CAM resin composites, except when repairing with a flowable resin composite. Despite the similar stress distribution among the repair composites, the flowable material presented less risk of failure, thus consisting of the most indicated choice for repair protocols.

## Figures and Tables

**Figure 1 dentistry-11-00254-f001:**
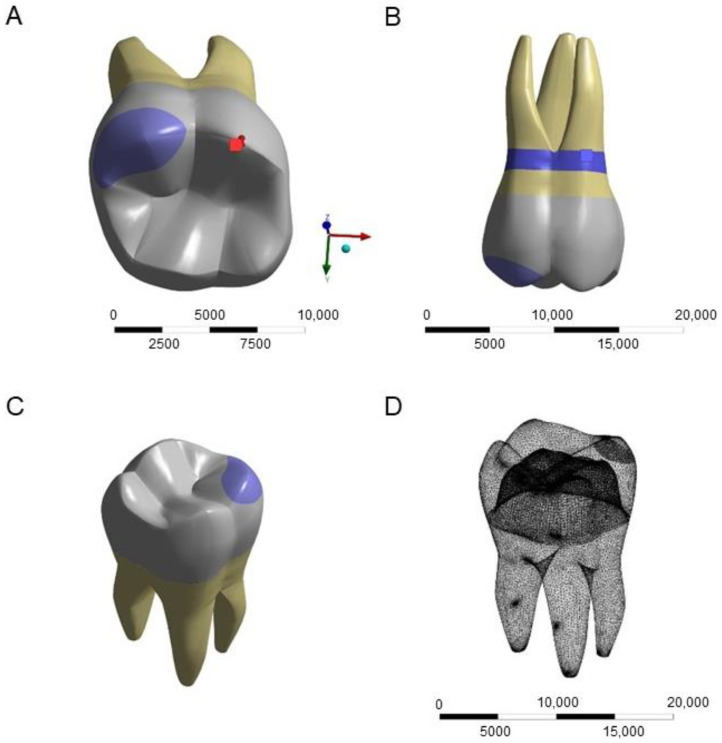
Study model presenting the cusp fracture of monolithic restorations. Occlusal view (**A**), buccal view (**B**), isometric view (**C**), and meshing (**D**).

**Figure 2 dentistry-11-00254-f002:**
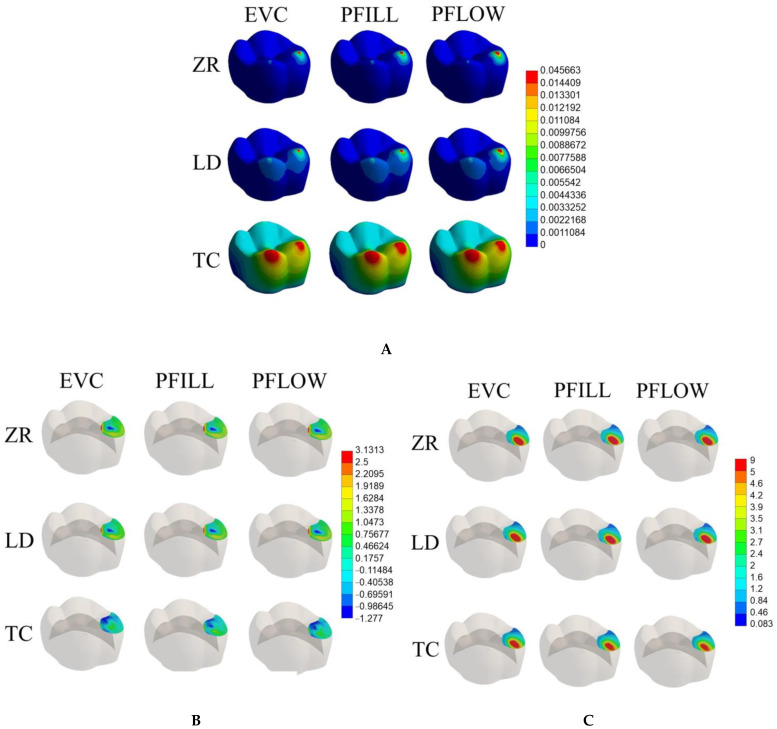
Total deformation per model according to different restorative materials and resin composite repairs (**A**). Tensile stress map per model according to different restorative materials and resin composite repairs (**B**). Shear stress map per model according to different restorative materials and resin composite repairs (**C**).

**Figure 3 dentistry-11-00254-f003:**
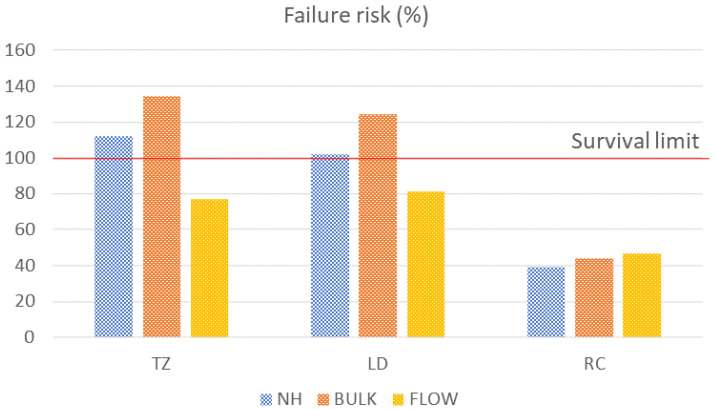
Risk of failure for each group. The red line represents 100% of the risk (survival limit). Translucent zirconia and lithium disilicate repaired with nanohybrid and bulk-fill resin composites trespassed the survival limit, while the CAD-CAM-resin-composite-repaired crowns presented the lowest failure risks.

**Table 1 dentistry-11-00254-t001:** Young modulus of the evaluated restorative and repair materials [8,19,20,21].

Materials	Young Modulus (GPa)	Poison Ratio
Lithium disilicate	95	0.3
Translucent zirconia	200	0.3
CAD-CAM resin composite	11.61	0.3
Nanohybrid repair composite	11	0.3
Bulk-fill repair composite	8.4	0.3
Flowable repair composite	5.6	0.3

(Özcan et al., 2020 [19]; Soares et al., 2021 [20]; Machry et al. 2022 [8]; Marovic et al., 2022 [21]).

**Table 2 dentistry-11-00254-t002:** Mean of tensile and shear peaks of stress through the finite element analysis. The mean (standard deviation) of shear bond strength through the in vitro shear test. Risk of failure (%) for each group.

Groups	Finite Element Analysis		In Vitro Test	Risk of Failure (%)
	Tensile Stress (MPa)	Shear Stress (MPa)	Shear Bond Strength (MPa)	
TZ-NH	3.09	8.97	7.99 (7.56) ^B^	112
TZ-BULK	3.13	9.00	6.69 (2.75) ^B^	135
TZ-FLOW	3.18	9.03	11.77 (5.44) ^AB^	77
LD-NH	2.90	8.81	8.66 (7.54) ^B^	102
LD-BULK	2.99	8.88	7.14 (4.13) ^B^	124
LD-FLOW	3.08	8.96	11.05 (3.16) ^AB^	81
RC-NH	1.77	7.03	17.96 (4.54) ^A^	39
RC-BULK	1.93	7.38	16.88 (8.15) ^A^	44
RC-FLOW	2.12	7.83	16.88 (5.26) ^A^	46

Different superscript letters indicate statistical differences among groups (α = 0.05) after the Tukey post hoc test.

## Data Availability

Data available on request.

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
