# Peer review of "Mechanical Behavior of Repaired Monolithic Crowns: A 3D Finite Element Analysis"

_dentistry, 2023, doi:10.3390/dj11110254_

Round 1

Reviewer 1 Report

Comments and Suggestions for Authors

The paper entitled “Mechanical behavior of repaired monolithic crowns: A 3D finite element analysis ” aimed to evaluate the mechanical behavior and risk of failure of different CAD-CAM materials (lithium disilicate, translucent zirconia, and resin composite) repaired with 3 classes of resin composite (nanohybrid, bulk-fill, and flowable resin), through a three-dimensional (3D) finite element analysis.

The paper covers an interesting topic. Nevertheless FEA analysis is quite a poor analysis method and  tensile shear should have been preferred to shear bond stregth.

The reviewer suggest major review:

Lines 54-7:

The authors wrote:

“To overcome the limitations of brittle crowns and to promote a homogenous stress distribution by using materials with similar  young modulus to those of dentin, CAD-CAM resin composite gained popularity for indirect restorations [7], with promising results regarding their mechanical behavior [8]. “

The authors should also mention the esthetic properties of CAD-CAM resin composites.

The authors could add a couple of citations related to esthetical and optical properties:

For lithium disilicate:

Comba A, Paolone G, Baldi A, et al. Effects of Substrate and Cement Shade on the Translucency and Color of CAD/CAM Lithium-Disilicate and Zirconia Ceramic Materials. Polymers (Basel). 2022;14(9):1778. Published 2022 Apr 27. doi:10.3390/polym14091778

For Composite cad/cam blocks:

Paolone G, Mandurino M, De Palma F, et al. Color Stability of Polymer-Based Composite CAD/CAM Blocks: A Systematic Review. Polymers (Basel). 2023;15(2):464. Published 2023 Jan 16. doi:10.3390/polym15020464

Line 128:

“and then light-cured for 10 s.”

please add type, manufacturer and light power (mW/cm2) of the curing unit

2.3. Statistical analysis 

How were the data normally analyzed?

The authors should mention the need (or not) of silane for composite repair (which should not be mandatory but…)

See: 

Dieckmann P, Baur A, Dalvai V, Wiedemeier DB, Attin T, Tauböck TT. Effect of Composite Age on the Repair Bond Strength after Different Mechanical Surface Pretreatments. J Adhes Dent. 2020;22(4):365-372. doi:10.3290/j.jad.a44867

 The authors should mention the need (or not) of a separate silane instead of the adhesive with included-silane

The authors should mention surface treatment differences: burs, air abrasion, silane, adhesive, curing or not the adhesive before repair.

The authors should also mention the clinical threshold of repair strength: meaning that lower repair strengths may be significantly lower than other procedures but still clinically acceptable.

Author Response

Reviewer 1:

The paper entitled “Mechanical behavior of repaired monolithic crowns: A 3D finite element analysis” aimed to evaluate the mechanical behavior and risk of failure of different CAD-CAM materials (lithium disilicate, translucent zirconia, and resin composite) repaired with 3 classes of resin composite (nanohybrid, bulk-fill, and flowable resin), through a three-dimensional (3D) finite element analysis.

The paper covers an interesting topic. Nevertheless FEA analysis is quite a poor analysis method and tensile shear should have been preferred to shear bond strength.

The reviewer suggest major review:

Answer: Thank you for your time to read our work and for the considerations that certainly will improve the quality of the study.

Lines 54-7:

The authors wrote:

“To overcome the limitations of brittle crowns and to promote a homogenous stress distribution by using materials with similar young modulus to those of dentin, CAD-CAM resin composite gained popularity for indirect restorations [7], with promising results regarding their mechanical behavior [8].“

The authors should also mention the esthetic properties of CAD-CAM resin composites.

The authors could add a couple of citations related to esthetical and optical properties:

For lithium disilicate:

Comba A, Paolone G, Baldi A, et al. Effects of Substrate and Cement Shade on the Translucency and Color of CAD/CAM Lithium-Disilicate and Zirconia Ceramic Materials. Polymers (Basel). 2022;14(9):1778. Published 2022 Apr 27. doi:10.3390/polym14091778

For Composite cad/cam blocks:

Paolone G, Mandurino M, De Palma F, et al. Color Stability of Polymer-Based Composite CAD/CAM Blocks: A Systematic Review. Polymers (Basel). 2023;15(2):464. Published 2023 Jan 16. doi:10.3390/polym15020464

Answer: Thank you. The sentence was reviewed and a reference was added to mention the aesthetic factor.

Line 128:

“and then light-cured for 10 s.”

please add type, manufacturer and light power (mW/cm2) of the curing unit

Answer: Reviewed. The requested information was added.

2.3. Statistical analysis 

How were the data normally analyzed?

Answer: Reviewed. Shapiro Wilk and Levene tests were performed to access the normality and homoscedasticity of the obtained data, respectively. This information was added to the manuscript.

The authors should mention the need (or not) of silane for composite repair (which should not be mandatory but…)

See: 

Dieckmann P, Baur A, Dalvai V, Wiedemeier DB, Attin T, Tauböck TT. Effect of Composite Age on the Repair Bond Strength after Different Mechanical Surface Pretreatments. J Adhes Dent. 2020;22(4):365-372. doi:10.3290/j.jad.a44867

Answer: The authors adopted the use of only an adhesive system to perform the conditioning of the resin composite crown to follow the manufacturer's recommendations and to use a compatible system with the repair materials used in the study.

 The authors should mention the need (or not) of a separate silane instead of the adhesive with included-silane.

Answer: The adopted adhesive system for the repair of the resin composite crown was chosen to follow the manufacturer's recommendations and to use a compatible system with the repair materials tested in the study. Besides, a previous study showed that the use of a silane agent before the adhesive was detrimental to the bond strength of resin composite repairs (Gutierrez et al., 2019).

References:

  • Gutierrez NC, Moecke SE, Caneppele TMF, Perote LCCC, Batista GR, Huhtalla MFRL, Torres CRG. Bond Strength of Composite Resin Restoration Repair: Influence of Silane and Adhesive Systems. J Contemp Dent Pract. 2019 Aug 1;20(8):880-886.

The authors should mention surface treatment differences: burs, air abrasion, silane, adhesive, curing or not the adhesive before repair.

Answer: Each surface treatment was performed based on the microstructure of the crown material to enhance the bonding to the repair resin composites. Besides, the surface treatment was not the factor under study, corroborating the use of the best-known protocol for each material, which is usually applied in a clinical scenario. This information was added to the discussion section.

The authors should also mention the clinical threshold of repair strength: meaning that lower repair strengths may be significantly lower than other procedures but still clinically acceptable.

Answer: Indeed, previous studies reported that repair procedures were effective in promoting longevity for dental restorations, as referenced in the introduction section. Besides, there is a study that evaluated the effect of the repair of endodontic access and aging on the fracture resistance of a zirconia restoration (Greuling et al., 2023). The study showed that the fracture resistance of a repaired zirconia was lower than the control group (without endodontic access/repair), however, that achieved strength was also satisfactory considering the clinical scenario.  

Reference:

  • Greuling, A., Wiemken, M., Kahra, C., Maier, H.J, Eisenburger, M. Fracture Resistance of Repaired 5Y-PSZ Zirconia Crowns after Endodontic Access. Dent J (Basel) 2023;11,76. https://doi.org/10.3390/dj11030076

Reviewer 2 Report

Comments and Suggestions for Authors

The fracture of the constructive material is not only determined by the mechanical properties of the material but also by a number of other factors. Therefore, the reader may be given distorted information.

Comments on the Quality of English Language

English is correct.

Author Response

The fracture of the constructive material is not only determined by the mechanical properties of the material but also by a number of other factors. Therefore, the reader may be given distorted information.

Answer: We agree. Several factors impact the mechanical behavior and longevity of dental restorations, such as the challenges of the oral environment, pH variations, parafunctional habits, type of load applied, among others. However, the purpose of the present study was to evaluate the stress distribution for different repaired restorations considering the influence of the material properties and microstructures as an isolated factor in a controlled in silico and in vitro scenario. Thus, it was possible to define the most predictable results when performing repair procedures in terms of material choice, which is an essential, even not definitive, step for achieving the best repair protocol for clinical applications. This limitation was added to the discussion.

Reviewer 3 Report

Comments and Suggestions for Authors

As the reviewer of the article titled "Mechanical Behavior of Repaired Monolithic Crowns: A 3D Finite Element Analysis," I have carefully assessed the various sections of the manuscript, including the Introduction, Materials and Methods, Results with Statistics, and the Discussion. This review offers an evaluation of the article, highlighting its strengths and areas for improvement.

Introduction:

The introduction is well-structured and effectively contextualizes the study within the broader field of dentistry. The authors provide a concise review of the relevant literature and clearly delineate the research gap their study addresses. The objectives and hypotheses are clearly defined, ensuring that readers can grasp the purpose of the investigation. The introduction is a strong foundation for the study, but it could be enhanced by incorporating additional recent references to further establish the context.

Materials and Methods:

The materials and methods section is meticulously presented, providing a comprehensive understanding of the experimental design. The detailed descriptions of CAD/CAM technology and the materials used are commendable, enhancing the study's reproducibility. The authors appropriately justify their choice of finite element analysis and define the boundary conditions. However, it would be beneficial for the authors to provide more information about the potential limitations and assumptions of their finite element analysis to ensure a deeper understanding of their methodology.

Results including Statistics:

The results section is one of the article's strengths. The presentation of numerical data, as well as the accompanying graphical representations, is clear and informative. The use of statistics is appropriate and contributes to the robustness of the findings. The results are well-structured and closely aligned with the research objectives. The authors have adhered to scientific conventions when presenting their findings, and the quality of the illustrations, such as figures and graphs, is commendable. It would be advantageous to include confidence intervals or error bars on relevant data points, enhancing the statistical rigor of the study.

Discussion:

The discussion section adeptly interprets the results within the context of the research objectives and prior literature. The authors provide insightful insights into the mechanical behavior of repaired monolithic crowns and discuss their findings meaningfully. The implications of the results for clinical practice are addressed, contributing to the practical significance of the research. To further enrich the article, it would be beneficial for the authors to expand on potential clinical applications and discuss the implications of their findings for the field of dentistry. Additionally, a more detailed discussion of the study's limitations would provide a more balanced perspective.

Bibliography:

The bibliography provides a reasonably comprehensive list of references that support the study. However, the authors could further enhance the article by including recent publications and expanding their review of relevant literature.

Overall Impression:

This article is a valuable addition to the field of dentistry, offering crucial insights into the mechanical behavior of repaired monolithic crowns. The study is well-structured and features high-quality graphics that improve the clarity of the presentation. The use of finite element analysis is justified and well-executed. The article exhibits high quality, and the minor revisions suggested in this review can further enhance its overall impact. I recommend accepting this article for publication with the condition that the authors address the minor revisions outlined above.

Author Response

As the reviewer of the article titled "Mechanical Behavior of Repaired Monolithic Crowns: A 3D Finite Element Analysis," I have carefully assessed the various sections of the manuscript, including the Introduction, Materials and Methods, Results with Statistics, and the Discussion. This review offers an evaluation of the article, highlighting its strengths and areas for improvement.

Answer: Thank you for your time to read our work and for the considerations that certainly will improve the quality of the study.

Introduction:

The introduction is well-structured and effectively contextualizes the study within the broader field of dentistry. The authors provide a concise review of the relevant literature and clearly delineate the research gap their study addresses. The objectives and hypotheses are clearly defined, ensuring that readers can grasp the purpose of the investigation. The introduction is a strong foundation for the study, but it could be enhanced by incorporating additional recent references to further establish the context.

Answer: Thank you for your consideration. We reviewed the references.

Materials and Methods:

The materials and methods section is meticulously presented, providing a comprehensive understanding of the experimental design. The detailed descriptions of CAD/CAM technology and the materials used are commendable, enhancing the study's reproducibility. The authors appropriately justify their choice of finite element analysis and define the boundary conditions. However, it would be beneficial for the authors to provide more information about the potential limitations and assumptions of their finite element analysis to ensure a deeper understanding of their methodology.

Answer: Reviewed. The limitations of the study at the end of the discussion section were added.

Results including Statistics:

The results section is one of the article's strengths. The presentation of numerical data, as well as the accompanying graphical representations, is clear and informative. The use of statistics is appropriate and contributes to the robustness of the findings. The results are well-structured and closely aligned with the research objectives. The authors have adhered to scientific conventions when presenting their findings, and the quality of the illustrations, such as figures and graphs, is commendable. It would be advantageous to include confidence intervals or error bars on relevant data points, enhancing the statistical rigor of the study.

Answer: Thank you for your consideration. The standard deviations and statistical differences are present for the in vitro phase of the study, as shown in Table 2. Regarding the finite element analysis, Since just one model was generated for each condition, we adopted a 10% mesh convergence test to validate the results, which was also used to evaluate the possible differences between the groups for the stress concentration, as described by Tribst et al. (2018).

Reference:

  • Tribst JPM, Dal Piva AMO, Penteado MM, Borges ALS, Bottino MA. Influence of ceramic material, thickness of restoration and cement layer on stress distribution of occlusal veneers. Braz Oral Res. 2018;32:e118.

 Discussion:

The discussion section adeptly interprets the results within the context of the research objectives and prior literature. The authors provide insightful insights into the mechanical behavior of repaired monolithic crowns and discuss their findings meaningfully. The implications of the results for clinical practice are addressed, contributing to the practical significance of the research. To further enrich the article, it would be beneficial for the authors to expand on potential clinical applications and discuss the implications of their findings for the field of dentistry. Additionally, a more detailed discussion of the study's limitations would provide a more balanced perspective.

Answer: Reviewed. The discussion and limitations were rewritten.

 Bibliography:

The bibliography provides a reasonably comprehensive list of references that support the study. However, the authors could further enhance the article by including recent publications and expanding their review of relevant literature.

Answer: Reviewed. Some references were replaced by recent ones.

 Overall Impression:

This article is a valuable addition to the field of dentistry, offering crucial insights into the mechanical behavior of repaired monolithic crowns. The study is well-structured and features high-quality graphics that improve the clarity of the presentation. The use of finite element analysis is justified and well-executed. The article exhibits high quality, and the minor revisions suggested in this review can further enhance its overall impact. I recommend accepting this article for publication with the condition that the authors address the minor revisions outlined above.

Answer: Thank you for your consideration and time to review our work.

Reviewer 4 Report

Comments and Suggestions for Authors

Well conducted study on stress on different prosthetic structures with CAD CAM technique.

Just a few criticisms listed below:

check that the keywords are Pubmed MESH terms

- insert some numerical references in the abstract results section

-in the initial part of the introduction section add some considerations on the various additive and subtractive CAD-CAM techniques that can be used in fixed prosthetics.

In this regard, I ask you to include the following scientific work in the reference section which could be of help to the reader:

Valenti C, Isabella Federici M, Masciotti F, Marinucci L, Xhimitiku I, Cianetti S, Pagano S. Mechanical properties of 3D-printed prosthetic materials compared with milled and conventional processing: A systematic review and meta-analysis of in vitro studies. J Prosthet Dent. 2022 Aug 5:S0022-3913(22)00415-2. doi: 10.1016/j.prosdent.2022.06.008. Epub ahead of print. PMID: 35934576.

- I prefer that the null hypotheses of the study are inserted at the end of the introduction section, which will then be refuted in light of the results obtained

-what is the rationale for choosing 100 N as the applied force? What media are you referring to? A bibliographical reference must be indicated regarding this

-a section on the limitations of the study is missing

-I ask for a general check on the English language in the manuscript

Comments on the Quality of English Language

-I ask for a general check on the English language in the manuscript

Author Response

Well conducted study on stress on different prosthetic structures with CAD CAM technique.

Answer: Thank you for your time to read our work and for the considerations that certainly will improve the quality of the study.

Just a few criticisms listed below:

check that the keywords are Pubmed MESH terms

Answer: Reviewed.

- insert some numerical references in the abstract results section

Answer: Reviewed.

-in the initial part of the introduction section add some considerations on the various additive and subtractive CAD-CAM techniques that can be used in fixed prosthetics.

Answer: Reviewed. Reference added to the manuscript.

In this regard, I ask you to include the following scientific work in the reference section which could be of help to the reader:

Valenti C, Isabella Federici M, Masciotti F, Marinucci L, Xhimitiku I, Cianetti S, Pagano S. Mechanical properties of 3D-printed prosthetic materials compared with milled and conventional processing: A systematic review and meta-analysis of in vitro studies. J Prosthet Dent. 2022 Aug 5:S0022-3913(22)00415-2. doi: 10.1016/j.prosdent.2022.06.008. Epub ahead of print. PMID: 35934576.

- I prefer that the null hypotheses of the study are inserted at the end of the introduction section, which will then be refuted in light of the results obtained

Answer: Reviewed. A null hypothesis was added.

-what is the rationale for choosing 100 N as the applied force? What media are you referring to? A bibliographical reference must be indicated regarding this

Answer: A previous study considering just zirconia as restorative material applied 200 N over the cusp of the molar models (Anami et al., 2015). A lower load was adopted considering the microstructure and mechanical properties of the restorative materials as resin composite and lithium disilicate, which present lower mechanical strength compared to zirconia. However, it must be considered that the stress distribution pattern is linear according to the response of each model after the load application. Thus, the differences in stress concentration would be the same after 100, 200, and 300N of loading.

Reference:

1- Anami LC, Lima JMC, Corazza PH, Yamamoto ETC, Bottino MA, Borges ALS. Finite element analysis of the influence of geometry and design of zirconia crowns on stress distribution. J Prosthodont. 2015 Feb;24(2):146-51. doi: 10.1111/jopr.12175.

-a section on the limitations of the study is missing

Answer: The limitations of the study are discussed at the end of the discussion section. Some new points were added.

-I ask for a general check on the English language in the manuscript.

Answer: The English was reviewed in the entire article.

Round 2

Reviewer 1 Report

Comments and Suggestions for Authors

The authors have provided all the requested improvements 

Reviewer 2 Report

Comments and Suggestions for Authors

The article does not provide valuable data.

Comments on the Quality of English Language

Minor editing is required.